# Comparative Genomic Analysis of Enterococci across Sectors of the One Health Continuum

**DOI:** 10.3390/microorganisms11030727

**Published:** 2023-03-11

**Authors:** Sani-e-Zehra Zaidi, Rahat Zaheer, Dominic Poulin-Laprade, Andrew Scott, Muhammad Attiq Rehman, Moussa Diarra, Edward Topp, Gary Van Domselaar, Athanasios Zovoilis, Tim A. McAllister

**Affiliations:** 1Lethbridge Research and Development Centre, Agriculture and Agri-Food Canada, Lethbridge, AB T1J 4B1, Canada; 2Department of Chemistry and Biochemistry, University of Lethbridge, 4401 University Drive, Lethbridge, AB T1K 3M4, Canada; 3Sherbrooke Research and Development Centre, Agriculture and Agri-Food Canada, Sherbrooke, QC J1M 1Z3, Canada; 4London Research and Development Centre, Agriculture and Agri-Food Canada, London, ON N5V 4T3, Canada; 5Guelph Research and Development Centre, Agriculture and Agri-Food Canada, Guelph, ON N1G 5C9, Canada; 6National Microbiology Laboratory, Public Health Agency of Canada, 1015 Arlington Street, Winnipeg, MB R3E 3R2, Canada

**Keywords:** comparative genomics, antimicrobial resistance, enterococci, livestock, One Health

## Abstract

Enterococci are Gram-positive bacteria that can be isolated from a variety of environments including soil, water, plants, and the intestinal tract of humans and animals. Although they are considered commensals in humans, *Enterococcus* spp. are important opportunistic pathogens. Due to their presence and persistence in diverse environments, *Enterococcus* spp. are ideal for studying antimicrobial resistance (AMR) from the One Health perspective. We undertook a comparative genomic analysis of the virulome, resistome, mobilome, and the association between the resistome and mobilome of 246 *E. faecium* and 376 *E. faecalis* recovered from livestock (swine, beef cattle, poultry, dairy cattle), human clinical samples, municipal wastewater, and environmental sources. Comparative genomics of *E. faecium* and *E. faecalis* identified 31 and 34 different antimicrobial resistance genes (ARGs), with 62% and 68% of the isolates having plasmid-associated ARGs, respectively. Across the One Health continuum, tetracycline (*tetL* and *tetM*) and macrolide resistance (*ermB*) were commonly identified in *E. faecium* and *E. faecalis*. These ARGs were frequently associated with mobile genetic elements along with other ARGs conferring resistance against aminoglycosides [*ant(6)-la*, *aph(3′)-IIIa*], lincosamides [*lnuG*, *lsaE*], and streptogramins (*sat4*). Study of the core *E. faecium* genome identified two main clades, clade ‘A’ and ‘B’, with clade A isolates primarily originating from humans and municipal wastewater and carrying more virulence genes and ARGs related to category I antimicrobials. Overall, despite differences in antimicrobial usage across the continuum, tetracycline and macrolide resistance genes persisted in all sectors.

## 1. Introduction

Antimicrobial resistance (AMR) is defined as the ability of the bacterial cell to avoid cell damage by antimicrobials [1]. Some bacteria are naturally resistant to certain antimicrobials through intrinsic or inherent traits. Antimicrobial resistance genes (ARGs) conferring intrinsic resistance are mostly passed through clonal inheritance and are rarely transferred within or among bacterial populations. However, some ARGs can be acquired and associated with mobile genetic elements (MGEs) including plasmids, transposons, and integrative and conjugative elements. These ARGs can be transferred to other bacteria through horizontal gene transfer [2] and thus contribute to the spread of AMR in different ecosystems [3]. Exposure of bacteria to antimicrobials can facilitate ARG acquisition and the proliferation of resistant populations within ecosystems [4]. In animal production, sub-therapeutic administration of antimicrobials through feed and water to treat or prevent infectious diseases is one example of a practice that can increase AMR. Indeed, the imposed selective pressure can exacerbate AMR in gut microbiomes as large numbers of bacterial members that carry ARGs on MGEs [5] may facilitate their dissemination, including transfer to pathogenic bacteria. Therefore, multiple organizations, including the Canadian Integrated Program for Antimicrobial Resistance Surveillance (CIPARS), European Antimicrobial Susceptibility Surveillance in Animals (EASSA), Japanese veterinary antimicrobial resistance monitoring systems (JVARM), and the National Antimicrobial Resistance Monitoring System for Enteric Bacteria (NARMS) in the United States are monitoring antimicrobial resistance in food animals and assessing their role in the dissemination of AMR to bacteria associated with humans.

*Enterococci* are commensal bacteria within the gastrointestinal tract of humans and animals [6]. They can also be recovered from broader natural environments, including soil, water, and plants. Some enterococcal species, particularly *Enterococcus faecalis* and *Enterococcus faecium*, are considered human pathogens as they are frequently associated with bacteremia, septicemia, meningitis, endocarditis, and urinary tract and wound infections [7]. The presence of *Enterococcus* spp. in different ecosystems makes them an ideal species to study AMR from a One Health perspective. We investigated the prevalence and nature of *Enterococcus* species recovered from swine feces and undertook a comparative analysis of *E. faecium* and *E. faecalis* genomes sourced across various sectors of the One Health continuum. More specifically, we evaluated (i) profiles of ARGs, MGEs, and virulence factors of these genomes, (ii) the association of MGEs with ARGs, and (iii) the phylogenetic relatedness of the isolates collected across different sectors.

## 2. Methodology

### 2.1. Enterococcus Recovery from Swine Feces and Whole Genome Sequencing

In 2017 and 2018, fecal samples were collected from sows, and weaning and finishing pigs raised on commercial antimicrobial-free farms, as well as conventional farms using penicillin prophylaxis in Quebec, Canada. Isolates were collected at the same time that *Enterobacterales* isolates were collected in a previous study [8]. Presumptive *Enterococcus* isolates were recovered from collected samples on Bile Esculin Azide (BEA) agar with and without erythromycin (8 µg/mL) as described previously [9] and a total of 41 isolates were confirmed to be *Enterococcus* species following PCR with Ent-ES-211-233-F and Ent-EL-74-95-R primers and Sanger sequencing of the PCR product [9]. Confirmed isolates were subjected to short-read Illumina sequencing. Genomic DNA was extracted using a Maxwell 16 Cell SEV DNA purification kit (Promega, Madison, WI, USA) as per manufacturer’s instructions, followed by DNA quantification using a Quant-it High-Sensitivity DNA assay kit (Life Technologies Inc., Burlington, ON, Canada). One nanogram of gDNA was used for genomic library construction using an Illumina NexteraXT DNA sample preparation kit and the Nextera XT Index kit (Illumina Inc., Vancouver, BC, Canada) according to manufacturer’s guidelines. All libraries were sequenced on an Illumina Miseq platform generating 2 × 300 base-paired end reads with a 600-cycle MiSeq reagent kit v3 (Illumina).

### 2.2. Collection of Enterococcus faecium and Enterococcus faecalis Genomes

A total of 622 *E. faecium* and *E. faecalis* genomes were included for comparative genomic analysis. These genomes originated from three sources: (i) swine isolates from this study (*n* = 18), (ii) a collection of genomes recovered from environmental and livestock isolates from Ontario (*n* = 66), and (iii) previously published data from poultry (*n* = 32) [10] and One Health continuum (*n* = 506) [9] studies. The number and the origin of *E. faecium* and *E. faecalis* genomes included in the analysis are summarized in Table 1. *E. faecium* and *E. faecalis* genomes were categorized into four groups/sectors based on their origin: (i) clinical, (ii) municipal wastewater, (iii) livestock, and (iv) environment.

### 2.3. Genome Assembly and Data Analysis

All enterococcal genomes included in this study were assembled de novo using the Shovill pipeline v.1.1.0 (https://github.com/tseemann/shovill accessed on 15 November 2022). Illumina adapters were removed using Trimmomatic v.0.36.5 [11]. All reads were then assembled de novo into contigs by SPAdes v.3.11.1 [12]. Assembly was evaluated by QUAST version 5.2.0 [13]. The contigs were then annotated using Prokka v.1.13.1 [14].

The annotated genomes were screened for the presence of antimicrobial resistance and virulence genes using ABRicate v.1.0.1 (https://github.com/tseemann/ABRICATE accessed on 20 November 2022) against the NCBI Bacterial Antimicrobial Resistance Reference Gene Database (NCBI BioProject ID: PRJNA313047) and the VirulenceFinder database (PMID: 34850947) [15], respectively. All contigs were screened for the presence of plasmids using Mob-recon version 3.0.0 (https://github.com/phac-nml/mob-suite accessed on 10 January 2023) [16].

*E. faecium* (*n* = 246) and *E. faecalis* (*n* = 376) genomes were used for comparative genomics (Table 1). The core-genome phylogenomic trees were constructed using the SNVphyl pipeline version 1.2.3. The phylogenetic tree was generated by aligning paired-end Illumina reads against the respective reference genomes of *E. faecalis* (strain ATCC 47077/OG1RF; CP002621.1) and *E. faecium* (strain DO; CP003583.1) using SMALT (version 0.7.5; https://sourceforge.net/projects/smalt/ accessed on 12 January 2023). The generated read pileups were then subjected to quality filtering (minimum mean mapping quality score of 30), coverage cut-offs (15× minimum depth of coverage), and a single nucleotide variant (SNV) abundance ratio filter of 0.75 to obtain a multiple sequence alignment of SNV-containing sites. This SNV alignment (with no SNV density filtering) was used to create a maximum likelihood phylogeny using PhyML version 3.0. The generated Newick file was visualized using Interactive Tree Of Life (iTOL) version 6 [17].

Additionally, for *E. faecium* genomes, a *groEL*-based tree was constructed to investigate whether the genomes could be assigned to previously described hospital (clade A) or community (clade B) clades [18]. The extracted *groEL* gene sequence was aligned with *E. faecium* strain 75 V68 (Clade A) and *E. faecium* strain 81 (Clade B) using MAFFT version 7.490. The analysis included the *E. hirae* R17 (accession CP015516.1) *groEL* gene as an outgroup. The maximum-likelihood tree was then created with IQTree version 2.1.4.

Multilocus sequence typing (MLST) was also used to study the population structure and evolution of bacterial species. *E. faecium* and *E. faecalis* sequence types were assigned through the MLST scheme of each respective species using PubMLST tool (http://cge.cbs.dtu.dk/services/MLST/ accessed on 15 January 2023) [19].

## 3. Results

### 3.1. Enterococci Recovered from Swine Feces

#### 3.1.1. Species Identification

Of the *Enterococcus* spp. recovered from fecal samples, 14 isolates were from sows, 15 isolates were from weaners, and 12 isolates were from finishers. Six different enterococcal species were identified [*E. hirae* (*n* = 15), *E. faecium* (*n* = 12), *E. faecalis* (*n* = 6)*, E. saccharolyticus* (*n* = 3)*, E. villorum* (*n* = 3), and *E. asini* (*n* = 2)] (Figure 1A).

#### 3.1.2. Genome Characterization

Across all isolates, 27 different ARGs/determinants were identified (Figure 1B). Overall, 39% of the identified enterococcal species were multidrug-resistant (MDR, resistant to ≥ 3 antimicrobials). MDR isolates were confined to three species: *E. faecalis* (67%), *E. hirae* (47%), and *E. faecium* (41%) (Table 2). The most common ARGs in *E. faecium*, *E. faecalis*, and *E. hirae* were associated with resistance to aminoglycoside (*aph(3′)-IIIa, ant(6)-Ia*)*,* tetracycline *(tetL, tetM),* macrolide (*ermB*)*,* and streptothricin (*sat4*) drug classes.

Nine out of the twenty-seven ARGs conferred intrinsic/inherent resistance, including *msrC* (100%), *eat(A)* (100%), and *aac(6′)-li* (41.6%) in *E. faecium*; *lsa(A)* (100%) and *dfrE* (100%) in *E. faecalis*; *aac(6′)-lid* (66.6%) in *E. hirae*; *dfrF* (100%) and *vanC*-operon (100%) in *E. saccharolyticus*; and *aac(6′)-Entco* (100%) in *E. villorum* and *E. asini.* The three genes, *aacA-ENT1, dfrG*, and *aacA-ENT2,* were only identified in *E. faecium* (16.6%)*, E. faecalis* (33.3%), and *E. hirae* (33.3%), respectively.

A total of 35 plasmids were identified in *Enterococcus* spp. [*E. faecalis* (*n* = 10), *E. faecium* (*n* = 12), and *E. hirae* (*n* = 13)] (Table 2). Among these, 11 plasmids harbored ARGs [*E. faecalis* (*n* = 2), *E. faecium* (*n* = 2), and *E. hirae* (*n* = 7)] (Table 2). A total of 34 and 13 virulence genes were identified *in E. faecalis* and *E. faecium,* respectively. Most virulence genes were associated with cytolysis, biofilms, and capsule formation (Table 2). The *E. faecium* core-genome phylogenetic tree formed two distinct clades, where all genomes except two recovered from sows and finishers, were found in one clade. *E. faecalis* also clustered into two clades, where one clade exclusively contained genomes from weaners. As for *E. hirae,* one clade contained all genomes except two isolated from finishers (Figure 1C).

### 3.2. Comparative Genomic Analysis of E. faecalis and E. faecium across the One Health Continuum

#### 3.2.1. Livestock Production

Comparative genomic analysis of *E. faecium* (*n* = 91) and *E. faecalis* (*n* = 81) collected from cattle, poultry, and swine was performed to investigate similarities and differences in the resistome, virulome, and mobilome profiles as well as the phylogenetic relatedness across the production sectors.

Overall, 48% of *E. faecium* genomes from livestock were MDR (resistant to ≥3 antimicrobials). Among livestock, *E. faecium* from poultry had the highest incidence of MDR (61%), followed by swine (50%) and beef cattle (43%) (Figure 2). Among *E. faecium* of bovine origin, two ARG profiles [(*ermB*, *tetL, tetM*) and (*ant(6)-Ia, spw, ermB, lnuB, lsaE, tetL, tetM*)] were the most frequent (Appendix A). Isolates harboring *dfrE* were frequently identified in all sectors. Two ARG profiles [(*dfrE, tetL, tetM*) and (*dfrE, ermB, tetL, tetM*)] were present in both swine and poultry, while one profile (*dfrE, ermB*, and *tetM*) was common to bovine and poultry isolates. Across livestock, chloramphenicol (*fexA* and *catA*) and oxazolidinone-resistant determinants (*optrA*) were exclusively found in *E. faecium* from cattle, whereas the *vanC*-operon was unique to poultry isolates. Aminoglycoside ARGs [*ant(6)-Ia, ant(9)-Ia, aph(3′)-IIIa*, and *spw*] were more prevalent in *E. faecium* isolated from poultry compared to other sectors (Figure 3A). In contrast, tetracycline ARGs (*tetL* and *tetM*) were found more frequently in *E. faecium* from cattle than those from poultry and swine. Moreover, *E. faecium* isolates from cattle and poultry shared similar ARGs associated with macrolide–lincosamide–streptogramin (MLS) resistance (*ermA*, *ermB*, *lnuB, lnuG, lsaG*, and *sat4*). In *E. faecium* from swine, only four ARGs associated with MLS resistance (*ermB, lsaG, mefA*, and *sat4*) were identified. Across livestock, *ermB* (57%) was most prevalent in isolates from cattle. In contrast, the trimethoprim-resistant determinant *dfrE* was found in all *E. faecium* genomes recovered from swine and 82.6% from poultry. Compared to other sectors, *drfE* and *dfrG* were infrequently associated with *E. faecium* isolated from cattle.

Mobilome analysis of *E. faecium* genomes showed that >60% of ARG-carrying plasmids were associated with isolates from cattle (Appendix A). Among these, pL8-A and pM7M2 were also found in poultry and swine isolates, respectively. MLST profiling identified 33 different genomic sequence types (STs) across the enterococci genomes, with 13 STs exclusive to beef cattle. In swine, only 3 STs were identified (ST94, ST133, ST272). In *E. faecium* from poultry, 10 STs were identified, with ST154 being the most common. None of the STs were shared across all livestock species (Appendix A).

The virulome of *E. faecium* did not vary across livestock species. The majority of virulence genes, including those responsible for biofilm formation (*bopD*, *clpC*, *clpP*), bile-salt hydrolysis (*bsh*), capsule formation (*cap8F*, *cpsA*, *cpsB*, and *hasC*), MSCRAMM-like proteins (*sgrA*), and pili formation (*srtC*) were found in >70% of the genomes of *E. faecium* from livestock. Two genes, *ebpA* and *lap* (encoding biofilm-associated pili), and a *Listeria* adhesion protein were identified in one poultry isolate (Appendix A).

Overall, 46% of *E. faecalis* were MDR with the highest incidence of MDR associated with isolates from dairy cattle (91%) followed by poultry (57%), swine (34%), and beef cattle (15%) (Figure 2). One ARG profile (*ermB, tetM, tetL*) was found across all livestock species (Appendix A). The ARG profile *ant(6)-Ia, aph(3′)-IIIa, ermB, tetL,* and *tetM* was present in 50% of poultry and 100% of *E. faecalis* genomes from dairy cattle. Similar to *E. faecium*, the oxazolidinone resistance gene (*optrA*) was occasionally (7% of genomes) present in *E. faecalis* isolated from cattle. The trimethoprim ARG (*drfE*) was mapped to 17% and 3% of *E. faecalis* isolates from swine and cattle, respectively, but was absent in poultry isolates. Chloramphenicol resistance profiles differed across sectors, as *catA8* was found in isolates from swine, whereas *catA7* was found in isolates from dairy cattle and *catA7* and *fexA* in isolates from beef cattle. Similarly, the profile of aminoglycoside ARGs also varied across livestock species. Aminoglycoside ARGs were most prevalent in isolates from dairy cattle, followed by poultry, swine, and beef cattle. Two ARGs, *ant(6)-la* and *aph(3′)-IIIa,* were prevalent across livestock species, whereas *aph(2″)-Ih* and *ant(9)* were unique to isolates from dairy and beef cattle, respectively. The ARG *str,* was found only in isolates obtained from beef cattle and poultry. Similarly, *aadE* was found only in isolates from swine and beef cattle. Tetracycline resistance determinants (*tetL* and *tetM*) were found in isolates across livestock sectors (Figure 4A; Appendix A).

Like *E. faecium*, plasmid profiling of *E. faecalis* found that 70% of isolates possessed plasmids that carried ARGs (Appendix A). Four ARG-carrying plasmids (DO plasmid, pCTN1046, p6742_2, pEf37BA, and pBC16) were found in both *E. faecium* and *E. faecalis*. Across livestock species, 29 STs were identified, with ST59 shared between swine, bovine, and dairy cattle isolates (Appendix A). Virulome profiles of *E. faecium* genomes were similar across livestock species (Appendix A). A total of 27 of the 39 virulence genes were mapped to several isolates collected across the livestock sectors (40–100% of genomes). Genes encoding cytolysin (*cylA, cylB, cylI, cylL, cylM, cylR1, cylR2*, and *cylS*) and the aggregation substance (*asa1*) were found in only one isolate from swine.

#### 3.2.2. One Health Continuum

Across the continuum, 35% of *E. faecium* were MDR, with the highest incidence of MDR found in clinical (CL) isolates (53%), followed by livestock (LS) (48%), municipal wastewater (MW) (23%), and environmental (EV) isolates (16%) (Figure 2). The ARG profile *dfrE, ermB,* and *tetM* was most common among MDR *E. faecium* from LS, EV, and MW (Appendix A). Aminoglycoside resistance genes were most prevalent in clinical genomes, followed by LS, MW, and EV (Figure 3A). Three aminoglycoside resistance genes, *ant(6)-Ia*, *aph(3′)-IIIa*, and *spw,* were found across the One Health continuum, with *ant(6)-Ia* and *aph(3′)-IIIa* being frequently mapped to plasmids (73% and 61%, respectively). These genes were found together in 91% of genomes. The bifunctional gene *aac(6′)-Ie/aph(2″)-Ia,* was found only in CL (5/36, 14%) and MW (3/56, 5.3%) isolates. Genomes harboring *aac(6′)-Ie/aph(2″)-Ia* were associated with five different plasmids (Appendix A). This gene was exclusively associated with an IS256 insertion element, except for one plasmid associated with IS6 and IS1216 in combination with *ermB* and *dfrG*. Chloramphenicol resistance was found in LS and MW isolates but not among those from other sources. The ARG *fexA* was associated with *Tn*554 on plasmid pFSIS1608820, and *catA* was mapped to two plasmids in MW isolates (Appendix A). ARGs conferring resistance to trimethoprim were more prevalent in CL, followed by MW, LS, and EV. Compared to CL isolates, where *dfrF* and *dfrG* were more prevalent, *dfrE* was found in EV, LS, and MW isolates. In all but one *dfrG*-positive genome, *fosX* was found in an antisense direction to *dfrG* at an intergenic distance of ~3.2 kb. Macrolide–lincosamides–streptogramin-resistant genotypes were prevalent in LS, followed by CL, EV, and MW.

Four ARGs conferring macrolide resistance (*ermA*, *ermB*, *ermT*, and *mefA*) were identified across the continuum. The ARG *ermB* was associated with plasmids 73% of the time. Moreover, in isolates from CL and LS, *ermB* along with the aminoglycoside ARGs *sat4, aph(3′)-IIIa*, and *ant(6)-la* were associated with *Tn*3 transposons. Similarly, *ermA* was also identified on plasmid pL8-A along with *ermB* and *ant(9)-la.* The ARG *ermA* was also found on plasmid pFSIS1608820 with *ant(9)-Ia, cfr, optrA, ermA, and fexA*. In contrast, *ermT* mapped only to plasmid p121BS. The lincosamide-resistant genes *lnuB* and *lsaE* were found together on 87% of plasmids. Glycopeptide resistance was found in clinical and poultry genomes, where *vanA* was found in pV24-3 and pF856 plasmids (Appendix A).

The core-genome-based phylogenomic tree of *E. faecium* formed two clades that were completely superimposed with the A and B clades identified by the *groEL* gene maximum-likelihood tree (Figure 3B). *E. faecium* genomes did not group based on sample source, except for the clinical isolates in clade A. Furthermore, clade A harboured more virulence genes and ARGs than clade B. Multilocus sequence typing of *E. faecium* genomes identified 72 different STs (Appendix A), with ST117 and ST17 being exclusive to human clinical isolates. Across the continuum, 37 virulence genes were identified, of which 15 were found in genomes from all sectors (Appendix A).

Overall, 40% of *E. faecalis* were MDR, with MDR isolates being most frequent in MW (51%) followed by LS (46%), EV (25%), and CL (32%) (Figure 2). Across all sectors, *ant(6)-Ia, aph(3′)-IIIa, ermB, tetL,* and *tetM* were frequently identified in MDR *E. faecalis* genomes (Appendix A). A total of 51 plasmids carrying one or more ARGs were identified (Appendix A). Among these plasmids, two were conjugative plasmids (related to AY855841 and CP028721), and two were identified as mobilizable plasmids (related to CP028286 and CP028836). Aminoglycoside ARGs were more prevalent in MW, followed by LS, CL, and EV (Figure 4A). Across all sectors, eight aminoglycoside ARGs were identified, with five (*ant(6)-Ia*, *aph(2″)-Ih, aph(3′)-IIIa,* and *str*) found in all sectors. Similar to *E. faecium*, *ant(6)-Ia* and *aph(3′)-IIIa* were frequently found together (61 genomes) and mapped to plasmids (71% and 75% of isolates, respectively). Chloramphenicol resistance genes were more prevalent in LS, followed by EV, CL, and MW. Five ARGs (*catA7*, *catA8*, *catP*, *cat-TC*, and *fexA*) were identified, with *catA7*, *catA8*, and *fexA* present in all sectors. These three genes were always associated with plasmids (Appendix A). Trimethoprim ARGs (*dfrF/G*) were identified more frequently in CL compared to other sectors, with *dfrF* found in >60% of CL genomes (19% on a plasmid). Across all sectors, MLS resistance was more prevalent in MW, followed by LS, CL, and EV. Three ARGs responsible for macrolide resistance (*erm A, ermB*, and *msr*) were identified, with *ermB* present in 60% of all genomes and frequently associated with plasmids (75%). One *ermB*-carrying plasmid, CP024844, was found exclusively in CL and MW genomes (40% *ermB*-positive isolates). Lincosamide ARGs were not found in EV genomes, whereas in CL genomes, only *lnuB* was identified. Tetracycline resistance was found more frequently in LS genomes, followed by EV, CL, and MW. Five different tetracycline ARGs were identified (*tetM, tetL, tetO, tetS,* and *tetW*), with *tetM* mapping to 76.5% of the genomes. Compared to *tetM* (18%), *tetL* (85%) was more frequently found on plasmids. Moreover, in 85% of *tetM*-positive plasmids, *tetL* was found together in close proximity with *tetM*. One *tetM-* and *tetL*-carrying plasmid, pS7316, was also prevalent in isolates from LS, CL, and EV. Oxazolidinone resistance ARGs were found only in EV and LS, which were more prevalent in EV than LS. In EV, two ARGs (*optrA* and *cfrC*) were identified, whereas in LS, only *optrA* was found.

Across the continuum, the core-genome-based *E. faecalis* phylogenomic tree formed two main clades, where one clade contained the majority of CW and MW genomes (Figure 4B). MLST profiling of *E. faecalis* identified 75 different STs (Appendix A), where 48 STs were source-specific (CL = 17, LS = 14, EV = 8, MW = 9). We identified 40 virulence genes across all *E. faecalis* genomes, with 28 shared across all sectors (Appendix A).

## 4. Discussion

Antimicrobial resistance is a serious concern for human and animal health and the global economy. One Health approaches to assess AMR recognize the role of multiple ecosystems in generating and spreading antimicrobial resistance genes [2]. In One Health studies, *Enterococcus* species have been used as ‘indicator bacteria’ to monitor ARG dissemination in ecosystems. In this study, we performed genomic characterization of *Enterococcus* species recovered from feces of weaners, finishers, and sows. Furthermore, we evaluated the ARGs identified in *E. faecium* and *E. faecalis* genomes across livestock and poultry production systems and cumulatively across the overall One Health continuum.

*E. hirae* was predominantly identified in swine feces, followed by *E. faecium* and *E. faecalis*. In studies from the US and Canada, *E. hirae* was frequently recovered from livestock [9,20]. In poultry, *E. faecium* has been isolated most frequently [21] and along with *E. faecium* and *E. faecalis* are often associated with human infections [9]. In all identified enterococcal species, tetracycline resistance determinants *tetL and tetM* were frequently found on the mobile plasmid pM7M2 (NC_016009). This plasmid has been previously identified in *E. faecalis* isolated from dairy cattle feces and was shown to transfer into *Streptococcus mutans* UA159 through natural transformation [22]. These findings show that these three *Enterococcus* spp. (i.e., *E. faecium*, *E. faecalis*, and *E. hirae*) can readily acquire ARGs in the gut micro-environment and possibly contribute to gene dissemination through plasmid-mediated ARG transfer.

We aimed to define the impact of differences in AMU across different livestock sectors on the occurrence of ARGs within enterococci. Across all livestock sectors, isolates from bovine sources had the lowest incidence of MDR, which may reflect the extent of antimicrobial usage in this livestock sector in Canada. According to the CIPARS 2019 report, most antimicrobials are administered to swine (<300 mg/PCU), followed by poultry (<200 mg/PCU) and cattle (<100 mg/PCU) (CIPARS, 2019). Regardless of the high MDR in poultry isolates, we did not find any isolates of poultry origin carrying ARGs conferring resistance to antimicrobials that were administered to poultry (Table 1). However, comparative genomics of enterococci identified that tetracycline and macrolide resistance genotypes were more prevalent in the beef production system compared to swine and poultry, a result that may reflect the greater use of these antimicrobials in beef cattle [23,24].

Mobile genetic elements play a significant role in gene dissemination within and across ecosystems. In our study, all ARGs, except those that were intrinsic, were mapped to plasmids in almost 80% *E. faecium* and *E. faecalis* isolates. Resistance to aminoglycosides, tetracyclines, trimethoprim, and MLS was identified across all ecosystems, with tetracycline and MLS being the most common. With these antimicrobials broadly used across sectors, the existence and persistence of resistant strains across the continuum is perhaps not surprising [25,26]. Their persistence may also be explained by the co-existence of these genes along with other ARGs, and other studies have found a strong association of tetracycline resistance ARGs (*tetL* and *tetM*) with other ARGs, including *ermB, ant(6)-la, aph(3′)-IIIa, lnu(G), lsaE*, and *sat4* [27]. These ARGs were often found on MGEs that may facilitate their spread in different ecosystems. Continuous exposure to one antimicrobial class in a particular ecosystem can also select for ARGs conferring resistance to other antimicrobial classes [28,29,30].

Some antimicrobial resistance determinants were found in some sectors but not others. For example, *aac(6′)-Ie/aph(2″)-Ia,* which is associated with high-level gentamicin resistance (HLGR), was only identified in *E. faecium* genomes from CL and MW. However, the association of this gene with MGEs may facilitate its spread to other human pathogens as it mapped to five different plasmids and was frequently associated with IS256 elements. Previously, *aac(6′)-Ie/aph(2″)-Ia* was associated with IS256 on the *Tn*5281 composite transposon in a conjugative pBEM10 plasmid in *E. faecalis* [31], with Tn4001 on plasmid pSK1 in *Staphylococcus aureus* [32], and Tn4031 in *Staphylococcus epidermidis* [33]. Glycopeptide-resistant genes *vanA* and *vanC* were identified in clinical and poultry isolates. The *vanA* operon was mapped to two plasmids in CL isolates, pV24-3 and pF856. Along with the *vanA*-operon, other ARGs (*ant(6)-Ia, aph(3′)-IIIa, ermB*, and *sat4*) were also mapped to pF856. This particular plasmid was first reported in a hospitalized patient associated with a vancomycin-resistant *Enterococcus* outbreak in Ontario, Canada [34].

Our phylogenomic analysis revealed a similar topology of *gro-EL*-based [35] and core-genome-based trees, with *E. faecium* segregating into two main groups. Our core-genome tree topology partitioned into two clades. In contrast, in a recent study by Sanderson et al. [36], clade B formed a paraphyletic clade rather than a monophyletic clade. Our findings also agree with previous studies [35,36], as more ARGs and virulence genes were associated with clade A than clade B isolates. Furthermore, most of the genomes associated with CL isolates clustered in clade A. Phylogenetically, *E. faecalis* genomes did not cleanly partition into clades by source and instead formed multiple clades that originated from multiple sources.

In conclusion, our study suggests that some resistant strains are universally present in all ecosystems, irrespective of antimicrobial pressure. However, some ARGs are exclusive to particular ecosystems, reflecting antimicrobial usage within that sector. Moreover, we also found that co-selection and association of ARGs with different MGEs likely facilitate the spread of ARGs across the One Health continuum. In addition, clinical *E. faecium* isolates formed a distinct cluster and were consistently mapped to a hospital associated clade.

## Figures and Tables

**Figure 1 microorganisms-11-00727-f001:**
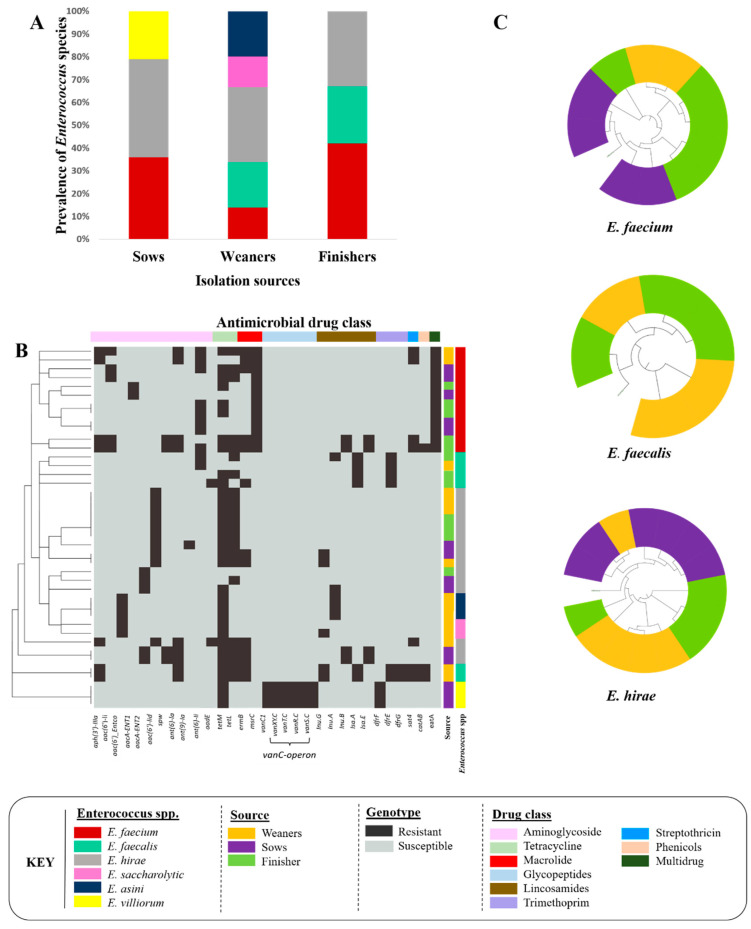
*Enterococcus* species recovered from fecal samples collected from sows (*n* = 14), and weaning (*n* = 15) and finishing (*n* = 12) pigs. (**A**) Prevalence of *Enterococcus* species. (**B**) Antimicrobial resistance gene profiles of *Enterococcus* isolates. (**C**) Core-genome-based phylogenetic tree of *E. faecium* (*n* = 12), *E. faecalis* (*n* = 6), and *E. hirae* (*n* = 15) recovered from different pig production stages.

**Figure 2 microorganisms-11-00727-f002:**
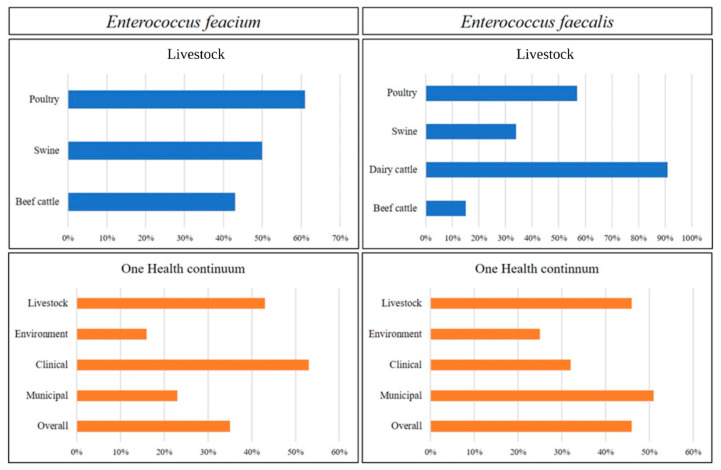
Multidrug resistant *Enterococcus faecium* and *Enterococcus faecalis* across One Health continuum.

**Figure 3 microorganisms-11-00727-f003:**
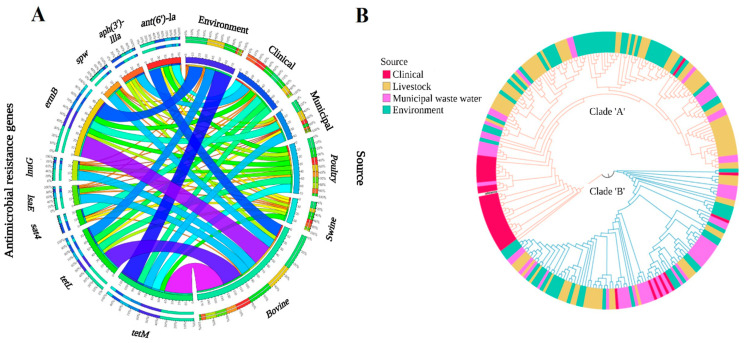
Comparative genomic analysis of 246 *E. faecium* genomes across the One Health continuum. (**A**) Circos plot depicts the relationship between commonly found ARGs and One-Health sectors. The variables (ARGs and genome isolation source) are arranged around the circle and distinguished by different colors. The percentage of ARGs across various sectors is indicated by proportional bars (http://circos.ca/). (**B**) Maximum likelihood core-genome phylogenetic tree. The *Enterococcus faecium* DO genome (CP003583.1) was used as a reference genome. The *gro-EL* gene-based *E. faecium* tree was overlaid on the core-genome *E. faecium* tree. Genomes were characterized based on their source of isolation into four groups: livestock, clinical, municipal wastewater, and environmental.

**Figure 4 microorganisms-11-00727-f004:**
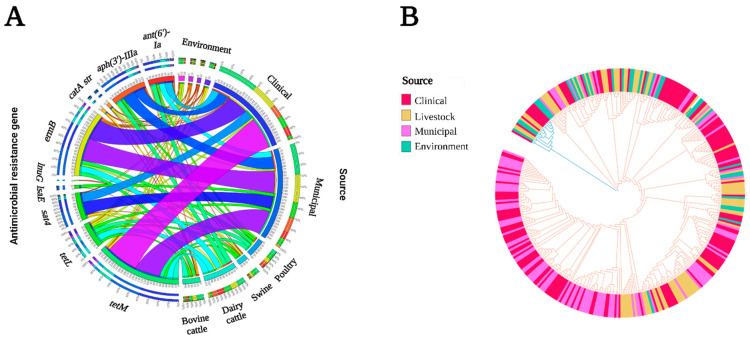
Comparative genomic analysis of 376 genomes *E. faecalis* genomes across the One Health continuum. (**A**) Circos plot depicts the relationship between commonly found ARGs and One Health sectors. The variables (ARGs and genome isolation source) are arranged around the circle and distinguished by different colors. The percentage of ARGs across various sectors is indicated by proportional bars (http://circos.ca/). (**B**) Maximum likelihood core-genome phylogenetic tree. *E. faecalis* ATCC 47077/OG1RF (CP002621.1) was used as the reference genome. Genomes were characterized based on their source of isolation into four groups: livestock, clinical, municipal wastewater, and environmental.

**Table 1 microorganisms-11-00727-t001:** Collection of *Enterococcus faecium* and *Enterococcus faecalis* genomes included in the comparative genomic analysis and antimicrobials used in livestock.

Sources of Genome		Number of Genome		Antimicrobial Usage	Location (Year of Sample Collection)	Reference
		*E. faecium* (*n* = 246)	*E. faecalis* (*n* = 376)			
**Municipal waste water (MW)**		56	110	-	Alberta (March 2014–April 2016)	[9]
**Clinical isolates (CL)**		36	149	-		
**Livestock (LS)**	**Bovine cattle**	57	33	Conventional (tetracycline, macrolides), natural (antibiotic-free)		
	**Dairy cattle**	-	22	NA	Ontario (2004)	This study
	**Swine**	-	06	NA		
		12	06	Conventional (penicillin), antibiotic-free (organic, certified-humane, AGRO-COM)	Quebec (2017–2018)	
	**Poultry**	23	09	Bambermycin, bacitracin, salinomycin, and β-lactams	British Colombia (2005–2008)	[10]
		-	05	NA	Ontario (2004)	This study
**Environment (EV)**	**Natural water sources**	46	19	-	Alberta (March 2014–April 2016)	[9]
	**River water**	16	07	-	Ontario (2004)	This study
	**Domestic animals**	-	03	NA		
	**Wild animals**	-	07	-		

**Table 2 microorganisms-11-00727-t002:** Antimicrobial resistance genes profiles, plasmids harboring AMR genes, and virulence genes identified in enterococcal species recovered from swine feces.

Enterococcal Species	*^&^ Antimicrobial Resistance Genes Profile (Number of Genomes)	Plasmids (Accession Number) (Total)	Antimicrobial Resistance Genes Found on Plasmid	Virulence Genes
** *E. faecalis* **	*aph(3′)-IIIa, ant(6)-la, tetL, tetM, ermB, lnu(G), dfrG, sat4, catA8* (*n* = 2)	pBEE99 (NC_013533) (*n* = 2)	All ARGs	Adhesive matrix molecules: *ace, fss1*, and *fss2* Biofilm formation: *bopD* Capsule formation: *cpsA-E* and *cpsG-K* Cytolysis: *cylA, cylB, cylI, cylL, cylM, cylR1*, *cylR2*, and cylS Endocarditis and biofilm-associated pili: *ebpA-C* and *srtC* Putative transporter protein: *efaA* Hyaluronidase: *EF0818* and EF3023) Gelatinase and serine protease: *fsrA-C, gelE,* and *sprE* Aggregation proteins: *prgB/asc10*
	*tetL, tetM* (*n* = 1)	pSWS47 (NC_022618.1) (*n* = 1)	All ARGs	
	*aadE, tetM, ermB* (*n* = 1)	None	None	
	*tetL, lnu(A)* (*n* = 1)	None	None	
** *E. faecium* **	*aph(3′)-IIIa, spw, ant(6)-Ia, tetL, tetM, ermB, lnu(B), lsa(E), sat4, catA8* (*n* = 1)	pM7M2 (NC_016009) (*n* = 4)	*tetL, tetM*	Adhesive matrix molecules: *acm, scm*, and *sgrA* Biofilm formation: *bopD, clpC, clpE*, and *clpP* Bile salt hydrolysis: *bsh* Capsule formation: *cap8F, cpsA, cpsB*, and *hasC* Pili formation: *srtC*
	*aph(3′)-IIIa, ant(6)-Ia, tetL, tetM, ermB, sat4* (*n* = 1)			
	*tetL, tetM, ermB* (*n* = 1)			
	*tetL, tetM* (*n* = 1)			
	*aph(3′)-IIIa, spw, ant(6)-Ia, tetL, tetM, ermB, lnu(B), lsa(E), sat4* (*n* = 1)	pLAG (KY264168.1) (*n* = 1)	*ant(6)-Ia, tetM, tetL, lnu(B), lsa(E)*	
	*aph(3′)-IIIa, ant(6)-Ia, ermB, sat4* (*n* = 1)	None	None	
	*tetM* (*n* = 3)	None	None	
** *E. hirae* **	*aph(3′)-IIIa, ant(6)-Ia, aadE, tetL, tetM, ermB, sat4* (*n* = 1)	p3 (CP006623) (*n* = 1)	*aph(3′)-IIIa, ant(6)-Ia, ermB, sat4*	Biofilm formation: *bopD* and *clpP* Hydrolysis of bile salt: *bsh*
		pBC16 (U32369) (*n* = 1)	*tetM*	
	*spw, ant(6)-Ia, tetL, tetM, ermB, lnuB, lsaE* (*n* = 2)	pEf37BA (MG957432) (*n* = 2)	All ARGs	
	*tetL, tetM, ermB, lnuG* (*n* = 2)	pDO1 (CP003584) (*n* = 2)	*tetL, tetM, ermB*	
	*ant(9)-Ia, tetL, tetM* (*n* = 1)	pM7M2 (NC_016009) (*n* = 1)	*tetL, tetM*	
	*tetL, tetM* (*n* = 7)	pM7M2 (NC_016009) (*n* = 3)	*tetL, tetM*	
		pCTN1046 (CP007650) (*n* = 1)	*tetM*	
		pBC16 (U32369) (*n* = 1)	*tetL*	
	*tetM, lnuA* (*n* = 1)	(CP029969) (*n* = 1)	*lnu(A)*	
** *E. asini* **	*tetM, lnuG* (*n* = 1)	None	None	Adhesion associated gene: *fss3*
	*tetM* (*n* = 1)	None	None	
** *E. villorum* **	*tetM, lsaA* (*n* = 3)	None	None	None
** *E. saccharolyticus* **	*tetM* (*n* = 3)	None	None	Adhesion associated gene: *fss3*

* Antimicrobial drug classes and resistance genes: aminoglycoside (*ant(9)-Ia, aph(3′)-IIIa, ant(6)-Ia, aadE, spw*); tetracycline (*tetL, tetM*); macrolide (*ermB*), lincosamide ARG (*lnuA, lnuG, lsaA, lsaE*), chloramphenicol (catA8), trimethprim (*dfrG*). ^&^ All ARGs except for those shown in column 4 were mapped onto chromosomes.

## Data Availability

The draft whole genome sequence assemblies of the *Enterococcus* spp. recovered from One Health sectors (clinical, bovine cattle, dairy cattle, swine, environment, municipal waste water) are available in GenBank under Bio Projects PRJNA604849. Whole genome sequence assemblies of *Enterococcus* spp. from poultry are available in GenBank under Bio Project PRJNA273513.

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
