# Peer review of "Comparative Genomic Analysis of Enterococci across Sectors of the One Health Continuum"

_microorganisms, 2023, doi:10.3390/microorganisms11030727_

Round 1
Reviewer 1 Report
In this manuscript, a genome-level comparison of enterococci strains isolated from different parts of the environment was performed.
The following comments are made:
1. The abstract says: “We undertook a comparative genomic analysis of the resistanceome, virulome and mobilome of 246 E. faecium and 376 E. faecalis recovered from swine, beef cattle, poultry, dairy cattle, human clinical samples, municipal wastewater and environmental sources”, and the introduction states: “The first aim of this study was to recover porcine isolates from swine feces and identify the prevalence of enterococcal species and their genomic characterization. Secondly, a comparative analysis was conducted on E. faecium and E. faecalis genomes sourced across various sectors of the One Health continuum”. It seems that they are two different aims. You could unify the aim in the two parts of the manuscript. To really know what the aim of your work is.
2. Lines 100-101, it is not clear how you confirmed that they are Enterococcus strains, what sequence you used for confirmation.
3. Line 113. Indicate how many genomes are from this study.
4. Line 115. Say where the samples are from iii) so as not to have to look in the references.
5. There are genomes of strains isolated from different regions of the country, how valid is the comparison?
6. Line 182. Put the type of antibiotic to which the genes correspond.
7. Table 2. Put at the foot of the table the type of antibiotic to which the genes correspond, in column two.
8. Figure 1. Explain what occurs in A, B and C.
9. Table 3, You could be put on a bar graph to better appreciate the percentages.
10. Figure 3. You could enlarge it to appreciate it better. Put the key of the colors of Figure 3a.
11. “More specifically, we evaluated (i) ARGs 88 profiles of these genomes and their linkage with antimicrobial usage in each 89 sector, (ii) MGEs associated with ARGs, and (iii) phylogenetic relatedness of 90 isolates collected across sectors”. What are the conclusions of your three objectives?
Author Response
Microorganisms– Reviewers’ comments
Zaidi et al. -- Comparative genomic analysis of enterococci across sectors of the One Health continuum (manuscript ID: 2235369)
Review(s)’ Comments to Author
Reviewer 1
- The abstract says: “We undertook a comparative genomic analysis of the resistome, virulome and mobilome of 246 E. faecium and 376 E. faecalis recovered from swine, beef cattle, poultry, dairy cattle, human clinical samples, municipal wastewater and environmental sources”, and the introduction states: “The first aim of this study was to recover porcine isolates from swine feces and identify the prevalence of enterococcal species and their genomic characterization. Secondly, a comparative analysis was conducted on E. faecium and E. faecalis genomes sourced across various sectors of the One Health continuum”. It seems that they are two different aims. You could unify the aim in the two parts of the manuscript. To really know what the aim of your work is.
As recommended the aim has been unified. Text has been changed accordingly to reflect the study objectives with more clarity. (Page 2, Lines 75-82)
- Lines 100-101, it is not clear how you confirmed that they are Enterococcus strains, what sequence you used for confirmation.
Required information along with reference (Zaheer et al. 2020) has been provided in methodology section (line 89-94) ‘Presumptive Enterococcus isolates were recovered from collected samples on Bile Esculin Azide (BEA) agar with and without erythromycin (8µg/mL) as described previously [9] and a total of 41 isolates were confirmed to be Enterococcus species following PCR with Ent-ES-211-233-F and Ent-EL-74-95-R primers and Sanger sequencing of the PCR product [9].
- Line 113. Indicate how many genomes are from this study.
Required information has been provided in methodology section (Page 3, line 107-111)
- Line 115. Say where the samples are from iii) so as not to have to look in the references.
Required information has been provided in methodology section (Page 3, line 109-110)
- There are genomes of strains isolated from different regions of the country, how valid is the comparison?
The aim of the study was to compare the antimicrobial usage and resistome of E. faecalis and E. faecium across one-health sectors. To investigate that on broader spectrum we had included samples from different livestock sectors including bovine cattle, poultry and swine. We had included samples from different provinces and those provinces are the leading producer in the food industry for that particular sector. For instance, bovine cattle samples were from Alberta as majority of cattle producing feedlots in Canada are located in this province and it is the leading beef producer in the country. Similarly, swine samples are from Quebec and this province is the leading pork producer in Canada. Poultry samples are from British Columbia and it is one the leading producing province in Canada. Furthermore, there is often a great deal of commonality in the nature of the livestock production systems across Canada, particularly for swine and poultry which are most often produced within confinement barns and fed similar diets regardless of where they are produced within Canada.
- Line 182. Put the type of antibiotic to which the genes correspond.
Required information has been provided in result section (page 7 line 175-179)
- Table 2. Put at the foot of the table the type of antibiotic to which the genes correspond, in column two.
Required information has been provided in footnote of table 2.
- Figure 1. Explain what occurs in A, B and C.
Detail information has been provided in figure 1 caption. Figure 1. Enterococcus species recovered from fecal samples collected from sow (n=14), weaning (n=15) and finishing (n=12) pigs (A) Prevalence of Enterococcus species (B) Antimicrobial resistance gene profiles of Enterococcus isolates (C) Core genome based phylogenetic tree of E. faecium (n=12), E. faecalis (n=6) and E. hirae (n=15) recovered across pig production stages.
- Table 3, You could be put on a bar graph to better appreciate the percentages.
Table 3 has been replaced with figure (Figure 2) as suggested.
- Figure 3. You could enlarge it to appreciate it better. Put the key of the colors of Figure 3a.
Figure enlarged as requested and description of colors added:
Figure 3a is the Circos plot that depicts the relationship between commonly found ARGs and One-Health sectors. The variables (ARGs and genome isolation source) are arranged around the circle and distinguished by different colors: The percentage of ARGs across various sectors is indicated by proportional bars (http://circos.ca/)
- “More specifically, we evaluated (i) ARGs 88 profiles of these genomes and their linkage with antimicrobial usage in each 89 sector, (ii) MGEs associated with ARGs, and (iii) phylogenetic relatedness of 90 isolates collected across sectors”. What are the conclusions of your three objectives?
Conclusion has been provided in discussion section (line 430-436) ‘In conclusion, our study suggests that some resistant strains are universally present in all ecosystems, irrespective of antimicrobial pressure. However, some ARGs are exclusive to particular ecosystems, reflecting antimicrobial usage within that sector. Moreover, we also found that co-selection and association of ARGs with different MGEs likely facilitate the spread of ARGs across the One Health Continuum. In addition, clinical E. faecium isolates formed a distinct cluster and were consistently mapped to a hospital associated clade’.

Reviewer 2 Report
The work presented here is well organized and presented in a very good manner
just need to correct some minor things, at beginning clarify your research question and research hypothesis in the introduction
also need to add some information about the previous research
Author Response
Reviewer 2
The work presented here is well organized and presented in a very good manner
just need to correct some minor things, at beginning clarify your research question and research hypothesis in the introduction
also need to add some information about the previous research
Thanks for your useful recommendations. Manuscript text has been modified to address these comments.

Reviewer 3 Report
Comments to Authors-
The topic of the manuscript is of interest. Authors performed comprehensive analysis to come out with significant inferences. However, there are some improvements required in the manuscript. I came out with some revisions. Therefore, I recommend that some improvements should be addressed. I will explain my viewpoints in more detail below.
Major Revisions-
Page 6, line 458, “GenBank under Bio Projects PRJNA604849 and PRJNA273513”. Are these Bio Projects referred from this study, if these Bio Projects are previously submitted, did the assemblies of this study have been submitted in these Bio Projects. Please simplify the submission of the assemblies.
Minor Revisions-
1. Page numbers in the manuscript should be corrected.
2. Page 4, line 136, “[pmid 15608208]”, not able to access the record. Please indicate the database name such as nucleotide at GenBank.
3. Please check the format of citations in the manuscript. For example, page 4, line 138, page 5, line 153.
4. Page 5, line 158. Please indicate, where these clades are illustrated in the manuscript?
5. Page 5, line 164, “MLST scheme”, which software was used to identify MLST.
6. Page 6, line 180, Figure 1. Caption needs to be expanded with more clarity that should explain each sub figures A, B and C explicitly. Figure 1A has space in between individual groups, please clarify or update the separation in the bar chart of an individual group. Figure 1C, represents only three species, however it has identified total of six, the explanation should be expressed briefly in the caption.
7. Page (not assigned), line 187, “vanC-operon (100%) in E. saccharolyticus, and aac(6')-Entco (100%) in E. villorum and E. asini”. However, in Figure 1B, vanC-operon is scaled at E. villorum. Please check it.
8. Page (not assigned), line 191, “A total of 36 plasmids”. Please double check the numbers, in table 2, it is different.
9. Page (not assigned), line 227, “Multidrug resistance Enterococcus faecium and Enterococcus faecalis”. Species name should be italic.
10. Page 3, line 268, “(Table S6).” It should be “Supplementary Table S6”. Please check the entire manuscript.
11. Figure 3. The resolution should be improved. It could be 600dpi.
12. Table 2. “DO plasmid 1”. Is 1 representing the total number of plasmid? Please double check.
13. Table 2. Column 2. Are these genes at Chromosomes? Please indicate it in the table.
14. Table 3. Format of the table should be improved. “6/12; 50%” should be in one line. The calculation (“44/91; 48%”) is also not clear since it is in the down line.
15. In supplementary files, empty cells should be replaced by hyphen ( - ).
16. Page 2, line 236, if possible, the percentages should be indicated in the supplementary files also. Similarly check other Supplementary files where the percentage can be included.
17. Double check the links in the manuscript, they are in different font sizes than the text. For example, page 4, line 134, line 138.
18. Page 7, References are not according to the journal's guideline. For example, “doi:” should not be included in the reference format. Besides, some references have journals’ names abbreviated, others are with full names. Please check entire format of references carefully.
Author Response
Reviewer 3
Major Revisions-
Page 6, line 458, “GenBank under Bio Projects PRJNA604849 and PRJNA273513”. Are these Bio Projects referred from this study, if these Bio Projects are previously submitted, did the assemblies of this study have been submitted in these Bio Projects. Please simplify the submission of the assemblies.
These BioProjects have been previously submitted and assemblies from this study have also been submitted. Required information has been provided on page 7 (line 456-460). ‘The draft whole genome sequence assemblies of the Enterococcus spp. recovered from one-health sectors (clinical, bovine cattle, dairy cattle, swine, environment, municipal waste water) are available in GenBank under Bio Projects PRJNA604849. Whole genome sequence assemblies of Enterococcus spp. from poultry are available in GenBank under Bio Project PRJNA273513’.
Minor Revisions-
- Page numbers in the manuscript should be corrected.
Corrected
- Page 4, line 136, “[pmid 15608208]”, not able to access the record. Please indicate the database name such as nucleotide at GenBank.
Corrected on line 136 (now line 129; PMID: 34850947) recent reference has been provided for VirulenceFinder (Liu B, Zheng D, Zhou S, Chen L, Yang J. VFDB 2022: a general classification scheme for bacterial virulence factors. Nucleic Acids Res. 2022 Jan 7;50(D1):D912-D917. doi: 10.1093/nar/gkab1107. PMID: 34850947; PMCID: PMC8728188.)
- Please check the format of citations in the manuscript. For example, page 4, line 138, page 5, line 153.
Corrected. All reference were updated as per journal requirement
- Page 5, line 158. Please indicate, where these clades are illustrated in the manuscript?
These clades are shown in Figure 3B. We constructed groEL based tree using MAFFT version 7.490. This tree depicted the formation of two clades (clade A and B) as indicated in the figure. In that analysis, groEL sequence of E. faecium strain 75 (Clade A representative; Hung, 2019) and V68 E. faecium strain 81 (clade B representative; Hung, 2019) were also included as reference controls for clades A and B. Additionally, phylogenomics was conducted using core-genomes. Our core-genome tree also formed two clades A and B and was in perfect sync with the groEL based tree. Therefore, clades A and B were labelled in the core-genome based tree (Figure 3B) according to our gro-EL based tree. A reference has also been provided in methodology section of the described protocol (Hung, 2019)
- Page 5, line 164, “MLST scheme”, which software was used to identify MLST.
Required information has been provided in methodology section (page 5 line 157). We used PubMLST tool and MLST scheme as described by Jolley et al. 2018.
- Page 6, line 180, Figure 1. Caption needs to be expanded with more clarity that should explain each sub figures A, B and C explicitly. Figure 1A has space in between individual groups, please clarify or update the separation in the bar chart of an individual group. Figure 1C, represents only three species, however it has identified total of six, the explanation should be expressed briefly in the caption.
- Required information has been provided in figure 1 caption. ‘Enterococcus species recovered from fecal samples collected from sow (n=14), weaning (n=15) and finishing (n=12) pigs (A) Prevalence of Enterococcus species (B) Antimicrobial resistance gene profiles of Enterococcus isolates (C) Core genome based phylogenetic tree of faecium (n=12), E. faecalis (n=6) and E. hirae (n=15) recovered across pig production stages.
- Figure 1A: clustered bar graph has been replaced with stacked bar to clarify proportion of each group.
- We intended to investigate if phylogenetic trees of Enterococcus spp. depicts any clusters based on isolation source (sows, weaners and finishers). faecium, E. faecalis and E. hirae were the only species that were recovered from more than two sources. The other three species were found in only one source and that’s why we did not perform phylogenetic analysis of those species.
- Page (not assigned), line 187, “vanC-operon (100%) in E. saccharolyticus, and aac(6')-Entco (100%) in E. villorumand E. asini”. However, in Figure 1B, vanC-operon is scaled at E. villorum.Please check it.
Correction has been made in Figure 1. E. villorum and E. saccharolyticus were labeled wrongly in key of figure 1 thanks for identifying this error.
- Page (not assigned), line 191, “A total of 36 plasmids”. Please double check the numbers, in table 2, it is different.
Corrected (on page 187)
- Page (not assigned), line 227, “Multidrug resistance Enterococcus faecium and Enterococcus faecalis”. Species name should be italic.
Corrected (line 229)
- Page 3, line 268, “(Table S6).” It should be “Supplementary Table S6”. Please check the entire manuscript.
Corrected (line 272)
- Figure 3. The resolution should be improved. It could be 600dpi.
All figures have been converted to 600dpi
- Table 2. “DO plasmid 1”. Is 1 representing the total number of plasmid? Please double check.
Corrected in Table 2. DO plasmid 1 is replaced with pDO1. There are three DO plasmids named pDO1, pDO2, pDO3. We identified pDO1 in our isolates.
- Table 2. Column 2. Are these genes at Chromosomes? Please indicate it in the table.
Footnote has been added in Table 2 ‘All ARGs except for those shown in column 4 were mapped on the chromosome’.
- Table 3. Format of the table should be improved. “6/12; 50%” should be in one line. The calculation (“44/91; 48%”) is also not clear since it is in the down line.
This table has been replaced with figure 2 as per suggestion by reviewer 1.
- In supplementary files, empty cells should be replaced by hyphen ( - ).
All empty cells were replaced by hyphens in supplementary files
- Page 2, line 236, if possible, the percentages should be indicated in the supplementary files also. Similarly check other Supplementary files where the percentage can be included.
Percentages were added in all supplementary files.
- Double check the links in the manuscript, they are in different font sizes than the text. For example, page 4, line 134, line 138.
Corrected
- Page 7, References are not according to the journal's guideline. For example, “doi:” should not be included in the reference format. Besides, some references have journals’ names abbreviated, others are with full names. Please check entire format of references carefully.
All reference have been rearranged as per journal requirement

Reviewer 4 Report
The paper entitled “Comparative genomic analysis of enterococci across sectors of the One Health continuum” reports data about the presence of different antibiotic resistance genomic factors by WGS. The paper is in general well written and I not fatal bias are present in the manuscript. In general, the paper reports a bulk of information and the final conclusion are in line whit the reported data. Anyway, are the authors sure about genomes belonged to E. faecium were correctly assigned? This is because Belloso Daza and co-authors (2021) showed how many E. faecium isolates should be reassigned to the group of E. lactis. Please verify!
Here just a suggestion the authors could consider:
Row 111: It would be possible reporting all the accession numbers of the genomes considered in the study in a supplemental table?
Author Response
Reviewer 4
The paper entitled “Comparative genomic analysis of enterococci across sectors of the One Health continuum” reports data about the presence of different antibiotic resistance genomic factors by WGS. The paper is in general well written and I not fatal bias are present in the manuscript. In general, the paper reports a bulk of information and the final conclusion are in line whit the reported data. Anyway, are the authors sure about genomes belonged to E. faecium were correctly assigned? This is because Belloso Daza and co-authors (2021) showed how many E. faecium isolates should be reassigned to the group of E. lactis. Please verify!
Thanks for your useful recommendations. We confirmed Enterococcus species through PCR with Ent-ES-211-233-F and Ent-EL-74-95-R primers and Sanger sequencing of the PCR product’. Required information along with reference (Zaheer et al. 2020) has been provided in methodology section (line 89-94)
Here just a suggestion the authors could consider:
Row 111: It would be possible reporting all the accession numbers of the genomes considered in the study in a supplemental table?
Required information has been provided in ‘Genome Sequence data availability’ section (page 15). The genomes used for the analysis were from two previously submitted BioProjects. ‘The draft whole genome sequence assemblies of the Enterococcus spp. recovered from one-health sectors (clinical, bovine cattle, dairy cattle, swine, environment, municipal waste water) are available in GenBank under Bio Projects PRJNA604849. Whole genome sequence assemblies of Enterococcus spp. from poultry are available in GenBank under Bio Project PRJNA273513’.

Round 2
Reviewer 1 Report
The aim in the abstract is still different from the one expressed in the introduction, you must match it. In the abstract you only say that you compare the genomes and in the introduction you already specify what you did for. Correct
Author Response
Reviewer 1
The aim in the abstract is still different from the one expressed in the introduction, you must match it. In the abstract you only say that you compare the genomes and in the introduction you already specify what you did for. Correct
Text has been modified to add further clarity to the relevant sentences in the abstract and the introduction sections.
Reviewer 3 Report
The authors have improved all the comments according to the reviewer's report.
Author Response
Thank you! We acknowledge that incorporating your suggestions has improved the quality of the manuscript.